Journal of Machine Learning Research 23 (2022) 1-10        Submitted 1/21; Revised 5/22; Published 9/22

# NeXtMarker: Contrastive Learning for Marker-Level Interpretability in Single-Cell Multiplex Imaging

**Simon Gutwein**[1,2,3,4,5,6]                         SIMON.GUTWEIN@CCRI.AT

**Daria Lazic**[1]                                     DARIA.LAZIC@EMBL.DE

**Thomas Walter**[4,5,6]                               THOMAS.WALTER@MINESPARIS.PSL.EU

**Sabine Taschner-Mandl**[1]                           SABINE.TASCHNER@CCRI.AT

**Roxane Licandro**[3]                                 ROXANE.LICANDRO@MEDUNIWIEN.AC.AT

[1] *St. Anna Children's Cancer Research Institute, Vienna, Austria*

[2] *TU Wien, Institute of Visual Computing and Human-Centered Technology, CVL, Vienna, Austria*

[3] *Medical University of Vienna, Biomedical Imaging and Image-guided Therapy, Vienna, Austria*

[4] *Centre for Computational Biology (CBIO), Mines Paris, PSL University, 75006 Paris, France*

[5] *Institut Curie, 75248 Paris Cedex, France*

[6] *INSERM, U900, 75248 Paris Cedex, France*

## Abstract

Understanding cell phenotypes and their spatial organization is crucial in multiplex imaging for spatial biology. Conventional analysis pipelines rely on extensive preprocessing, including background correction and segmentation, introducing biases and information loss. We present NeXtMarker, an interpretable deep learning framework for end-to-end single-cell analysis of multiplex images, eliminating the need for manual preprocessing or segmentation. NeXtMarker employs learned marker-specific normalization and interpretable feature extraction to generate biologically meaningful embeddings in a fully self-supervised manner. It directly processes raw images of cells while preserving spatial and morphological information. We demonstrate NeXtMarker's ability to (i) resolve intercellular expression patterns and cell morphology, (ii) enable accurate cell phenotyping in a large neuroblastoma tumor dataset, and (iii) generalize to independent osteosarcoma images. NeXtMarker maintains high agreement with conventional pipelines while eliminating the need for preprocessing and segmentation and enhancing interpretability. By enabling unbiased, scalable single-cell analysis, NeXtMarker establishes a foundation for improved phenotyping in multiplex imaging. Code and pretrained models available at: [`code_released_upon_acceptance`].

**Keywords:** Multiplex Imaging , Spatial Biology, Single-Cell Analysis, Deep Learning, Interpretability

## 1 Introduction

Multiplex imaging (MI) technologies on the protein level, such as Imaging Mass Cytometry (IMC) (Giesen et al., 2014), Multiplexed Ion Beam Imaging (Angelo et al., 2014), and CO-Detection by Indexing (Black et al., 2021), enable the simultaneous detection of multiple biological markers while preserving spatial information. However, analyzing these high-dimensional images remains challenging (Bussi and Keren, 2024). Identifying and localizing cell phenotypes is essential for studying cell-type interactions in spatial biology and requires integrating both morphology and marker co-expression patterns. Conventional

phenotyping pipelines extract single-cell information through background correction, segmentation, and integrated expression (IE) quantification within cell segmentation masks (Fig. 1, A).

Despite widespread use in tools like Steinbock (Windhager et al., 2023), these pipelines have limitations: (i) Background correction relies on subjective user-defined annotations, introducing bias (Berg et al., 2019). (ii) Segmentation remains error-prone (Stringer et al., 2021; Greenwald et al., 2022), as low resolution and densely packed cells obscure cell borders (Bai et al., 2021), even after fine-tuning. (iii) IE assumes perfect segmentation and collapses complex image information into single-marker intensity values, disregarding intercellular expression patterns and morphology crucial for subtype differentiation. Additionally, manually defined features, such as IE, are not necessarily optimal for capturing the biologically most relevant information, potentially limiting the accuracy of phenotyping.

To address these limitations, we introduce NeXtMarker—a deep learning framework designed for unbiased exploration of cell types and states directly from raw multiplex imaging data, without requiring prior knowledge, annotations, or predefined cell categories. NeXtMarker eliminates biased, labor-intensive preprocessing, bypasses segmentation, and trains in a fully self-supervised manner. Operating directly on raw images, it enables large-scale analysis while preserving the spatial and morphological context. Its interpretable architecture reveals individual marker contributions, enhancing biological insight. Our key contributions with NeXtMarker are as follows:

(1) **No Preprocessing:** NeXtMarker analyzes raw multiplex images without background correction, segmentation, or manual preprocessing.

(2) **Interpretable Cell Embeddings:** The proposed architecture enables experts to assess marker contributions, revealing biologically relevant co-expression patterns.

(3) **Spatial and Morphological Preservation:** Unlike segmentation-based methods, NeXtMarker retains spatial marker variations and cell morphology for precise cell type characterization.

(4) **Normalization Learning:** The model learns marker-specific normalization, facilitating data integration and reducing batch effects.

## 2 Methods

NeXtMarker comprises three key components: (**i**) marker specific normalization learning, (**ii**) interpretable feature extraction, and (**iii**) feature embedding, capturing marker interactions, as shown in Fig. 1B. NeXtMarker processes image patches centered on individual cells and generates two outputs: an interpretable marker attribution vector, referred to as Interpretability stage ($\mathbf{I}$), indicating marker importance, and a final embedding vector ($\mathbf{F}$) used for downstream tasks, such as clustering. These outputs enable accurate, biologically interpretable cell phenotyping.

(**i**) **Normalization Learning**: Unlike integrative imaging methods that normalize intensities to [0,1], IMC operates on an unbounded intensity scale, with pixel intensities ranging

from 0 to over $10^3$, varying significantly between individual biological markers.. To standardize these variations from $[0, \infty)$ to $[0, 1]$, we apply parameterized sigmoid functions $f_M(x)$ per marker $M \in [M_1, ..., M_\mathbf{X}]$ (Fig. 1B, i). Each sigmoid function has two trainable parameters per marker: $x_{c,M}$, the center, and $x'_{c,M}$, the slope at $x_{c,M}$, learned and optimized during NeXtMarker's training. The transformation $f_M$ for marker $M$ and input pixel intensity $x$ is defined as:

$$f_M(x) = \frac{1}{1 + \exp\left(-4x'_{c,M} \cdot (x - x_{c,M})\right)} \tag{1}$$

**(ii) Interpretable Feature Extraction:** Deep learning models inherently lack interpretability unless explicitly designed for it, a critical factor for phenotyping in MI. Thus, we designed a feature extractor that disentangles features by input channel, corresponding to biological markers (Interpretability I in Fig. 1B, ii).

This is achieved via grouped convolutions (Xie et al., 2016), where the number of groups matches the number of markers ($G = \mathbf{X}$), ensuring each convolutional group learns features specific to a single marker. Stacking multiple such blocks forms a deep feature extractor that preserves marker-specific representations, enabling direct interpretation of biological marker contributions within the learned feature space.

**(iii) Channel Crosstalk**: Cell phenotyping relies on the simultaneous expression of specific marker combinations, hereafter referred to as co-expression. The final stage integrates

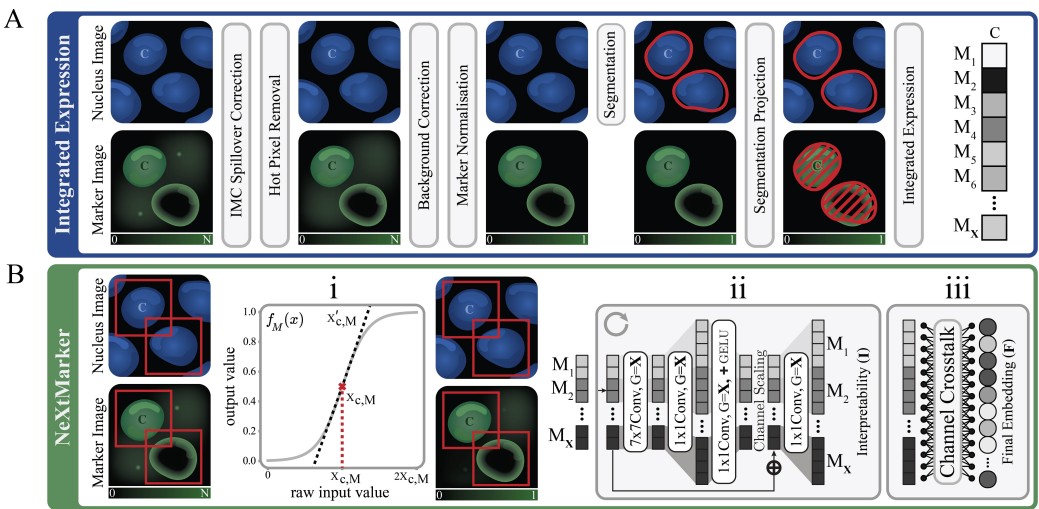

Figure 1: Overview of conventional analysis pipelines (A) compared to NeXtMarker's workflow (B). (**A**) Traditional pipelines involve preprocessing steps such as background correction, segmentation, and IE calculation. (**B**) NeXtMarker processes raw multiplex images without manual preprocessing, utilizing (**i**) a novel marker-specific normalization learning strategy, (**ii**) interpretable feature extraction, and (**iii**) crosstalk integration to generate feature embeddings.

marker interactions via a linear layer processing the Interpretability Vector I, producing the entangled final embedding $F$ (Fig. 1B, iii). This embedding refines feature organization and supports downstream tasks.

**Training of NeXtMarker:** NeXtMarker is trained in an self-supervised fashion using contrastive learning with a modified SimCLR (Chen et al., 2020) framework. Augmentations include random intensity scaling per marker, affine transformations, and flipping. Unlike standard SimCLR, which uses two positive pairs, we allow a flexible number of augmentations per sample. The model is optimized with NT-Xent (Chen et al., 2020) loss (temperature = 0.5). To stabilize learned normalization functions, we constrain parameter values, preventing negative $x_c$ or excessively steep $x'_c$, ensuring biologically plausible transformations using the following penalty term:

$$\mathcal{L}_M = \underbrace{\max(0, -x_{c,M})}_{Ensures\,x_{c,M} \geq 0} + \underbrace{\max(0, x'_{c,M} - 1)}_{Constrains\,x'_{c,M} \leq 1} + \underbrace{\max(0, 2/x'_{c,M} - x_{c,M})}_{Controls\,scaling\,relation} \tag{2}$$

## 3 Experiments and Results

NeXtMarker is evaluated through two experiments: (1) a synthetic dataset is used to test NeXtMarker's ability to distinguish intercellular expression patterns and morphology, and (2) two independent datasets are used with the same IMC marker panel, to test a real-world application and demonstrate NeXtMarker's effectiveness in cell phenotyping and cross-dataset generalization.

**Experiment 1 - Intercellular Expression Patterns and Morphology:** This experiment assesses NeXtMarker's ability to resolve intercellular expression patterns, which conventional IE methods fail to capture. We synthetically generated a five-marker $[M_1, ..., M_5]$ single-cell dataset with seven subpopulations, each defined by distinct marker expression profiles, cell size (Fig. 2A,I), morphology (circular vs. neutrophil, Fig. 2A,II), expression localization (center vs. border, Fig. 2A,III), and relative abundance.
NeXtMarker was trained with four augmented views per single cell patch, applying random marker intensity scaling (0.9 - 1.1), size scaling (0.9 - 1.1) rotation (0° - 359°), and random flipping (p=0.5). Training used the Adam optimizer (learning rate: 0.001, batch size: 256). For the baseline, we computed IE as the mean intensity over the nucleus mask per marker. NeXtMarker's performance was evaluated qualitatively via UMAP (McInnes et al., 2020) and quantitatively using a two-layer MLP classifier (60%-20%-20% train-validation-test split) for classification accuracy. Subpopulation separation was assessed with silhouette score (Rousseeuw, 1987) and Davies-Bouldin index (Davies and Bouldin, 1979).

**Results:** Fig. 2B shows that while IE distinguishes broad expression patterns, it fails to capture intercellular variations. In contrast, NeXtMarker effectively differentiates both expression patterns and subtypes, as confirmed by the confusion matrix (Fig. 2B) and quantitative metrics (Fig. 2C). Additionally, NeXtMarker organizes its feature space to reflect cell size variations within subtypes, evident in Fig. 2D, where a clear gradient from small to large cells is observed. This demonstrates NeXtMarker's ability to detect subtle morpho-

logical and marker expression differences.

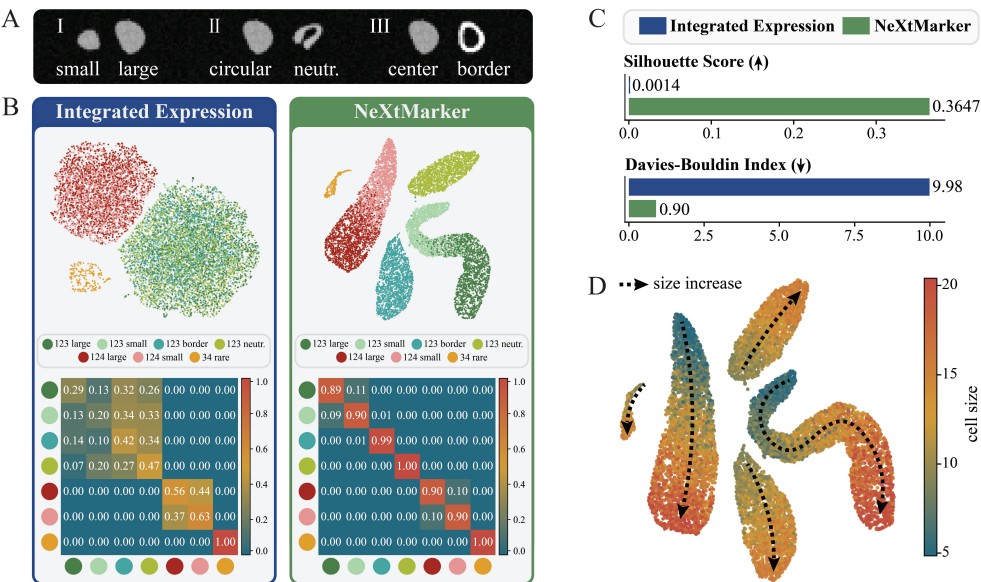

Figure 2: **A**: Synthetic images of subpopulations with distinct marker expression, morphology, and localization. **B**: UMAP visualization of Integrated Expression vs. NeXtMarker's feature space and subpopulation classification accuracy. **C**: Cluster separation metrics for both methods. **D**: UMAP of NeXtMarker's feature space colored by cell size.

**Experiment 2 - Real-World Phenotyping:** We evaluate NeXtMarker on IMC images from two cancer and tissue types: (I) 591 images from 144 bone marrow neuroblastoma samples (2002–2022) containing approx. 1.6 million cells and (II) a lung metastasis tissue slice from an osteosarcoma patient using the same IMC panel. The 34-marker panel (see Fig. 3A, y-axis) distinguishes immune and tumor cell types. Immune markers include those for T cells (CD3, CD4, CD8, GZMB), B cells (CD20), granulocytes (CD15), monocytes (CD14, CD10), dendritic cells (CD11), and progenitors (CD34, CD24). Tumor-associated markers include GD2, CD56, GATA3, SOX10, and EVALV4, enabling distinction of granulocytes, monocytes/dendritic cells/NK cells (MO/DC/NK), T cells, B cells, progenitors, and tumor cells. Cells without clear marker signatures are grouped as "others". NeXtMarker is benchmarked against the IE approach, hereafter referred to as the IE baseline, following the workflow in Fig. 1A. To minimize segmentation errors for the baseline, we used matched immunofluorescence nuclear scans at five times higher resolution for segmentation. For NeXtMarker, patch centers were estimated from the flow output of Cellpose (Stringer et al., 2021), without requiring full segmentation masks. To enable fair evaluation, each NeXtMarker center was mapped to its nearest segmentation-based cell. In cases where multiple cells were closest to the same center, the one with the smallest distance was assigned. Unmatched centers were excluded. NeXtMarker is not limited to this extraction method

and remains compatible with simple cell detection strategies, such as blob detection, local maxima in nuclear channels, or centroid extraction after thresholding.

**Cell Type Phenotyping Workflow:** In the IE workflow, feature vectors derived from averaged marker intensities over segmentation masks are clustered using PhenoGraph (Levine et al., 2015), and cluster-wise average marker expression is used for expert phenotype assignment. NeXtMarker follows a similar workflow but operates on its learned embeddings. After contrastive training, we extract interpretability (I) and embedding (F) vectors, cluster embeddings with PhenoGraph, and compute an average interpretability vector per cluster by averaging marker-associated features. This vector, capturing marker importance and co-expression, informs expert annotation. NeXtMarker's phenotype assignments are then compared to the IE pipeline for performance evaluation.

**Training Details:** NeXtMarker was trained using SimCLR using the following augmentations: intensity scaling (0.5 - 2), size scaling (0.66 - 1.5), rotations (0° - 359°), and random flipping (p=0.5). The learning rate was ramped up to 0.001 over 100 steps, followed by 1000 steps of cosine annealing and 100 steps at a constant rate at 0.0001. Training used a batch size of 1024 with two views (patch sizes 10, 8, and 6), exposing the model to $7 \times 10^6$ patches during training. The interpretability stage produced eight features per marker, with a final embedding dimension of 256. Normalization parameters were initialized with $x_c$ set to half the $90^{th}$ percentile of nonzero pixels per marker, and $x'_c = \frac{2}{x_c}$.

**Results:** Fig. 3A presents the average marker activation per cluster from NeXtMarker's interpretability stage (I), with color indicating activation levels and size representing the proportion of cells exceeding a predefined threshold. Fig. 3B shows the UMAP visualization of feature space and cluster assignments. We highlight selected columns in blue in Fig. 3A. With these examples we illustrate how key immune cell types are identified based on marker expression patterns: Cytotoxic T cells (clusters 27, 18, 34, 52) express CD3 and CD8, with or without GZMB, while T-helper cells (clusters 60, 3) show CD3 and CD4. NK cells, defined by GZMB without CD3/CD8, correspond to clusters 62, 42, and 25. These highlights provide a clear reference for understanding how NeXtMarker enables phenotyping in high-dimensional MI data without segmentation or manual feature engineering by leveraging learned interpretable features.

**Benchmarking Against Integrated Expression:** To compare NeXtMarker with the IE baseline, we project cell-type assignments from both methods onto NeXtMarker's UMAP feature space (Fig. 3C, left for IE baseline, center for NeXtMarker). Agreement is shown in Fig. 3C, right, where correctly matched annotations appear in green and mismatches in red. NeXtMarker achieves 82.07% overall agreement, with a detailed breakdown per cell type using a confusion matrix. Agreement is highest for B cells, MO/DC/NK cells, T cells, granulocytes, and tumor cells, while progenitors and the 'others' category remain challenging. Progenitors are often misclassified as granulocytes (24.3%) explainable due to overlapping morphology and marker expression. Similarly, 'others' are frequently assigned to granulocytes (24.9%) or progenitors (35.9%). While cells assigned as 'others' cluster closely together in feature space (Fig. 3G), they lack clear separation from other cell types, causing misclustering. To contextualize these results, we reran the IE pipeline with re-

trained background correction and clustering to assess its self-consistency. This yielded a similar overall agreement of 81.06% (Fig. 3D, right). The confusion matrix in Fig. 3D confirms similar misclassification trends, with progenitors frequently labeled as MO/DC/NK cells (14.2%) or granulocytes (9%), while 'others' were primarily assigned to progenitors (44.4%), MO/DC/NK cells (15.9%), or granulocytes (22.9%). These findings highlight the inherent ambiguity in phenotyping cells near class boundaries. One notable discrepancy in the baseline comparison is cluster 30, classified as B cells by NeXtMarker along with cluster 11, which has a similar expression profile. However, the IE baseline primarily assigned non-B cell labels to cluster 30. Single-cell patches (Fig. 3E) show that raw IMC images contain CD20 signal (B cell marker) (Fig. 3E, left), but the IE baseline's background correction removed most of this signal (Fig. 3E, right), leading to misclassification. In contrast, cells from cluster 11 (Fig. 3F) retained well-defined signals and were classified as B cells by both methods, suggesting that background correction may introduce phenotyping errors.

These findings demonstrate that NeXtMarker accurately identifies cell phenotypes without preprocessing or segmentation. By avoiding biases from background correction, it not only matches but, surpasses traditional methods, underscoring its potential for robust phenotyping.

**Cross-Dataset Generalization:** To assess NeXtMarker's ability to extract biologically meaningful features across datasets, we applied the on neuroblastoma trained model to an unseen IMC image of an osteosarcoma lung metastasis using the same IMC panel. A classifier was trained on the neuroblastoma dataset, using only cells whose phenotype labels matched their 10 nearest neighbors in feature space, then applied to the osteosarcoma image. As a case study, we focused on T cells due to their distinct marker profile and presence across tissues and diseases. Fig. 3H shows a strong correlation between CD3 or CD3+CD8 expression and T cell classification, consistent with known marker biology. This demonstrates that NeXtMarker captures biologically relevant relationships beyond dataset-specific conditions, enabling robust generalization across datasets and tissue types.

## 4 Conclusion

We introduced NeXtMarker, an interpretable deep learning framework for single-cell analysis of multiplex imaging data, eliminating the need for manual preprocessing and segmentation. By learning marker-specific normalization, disentangled feature extraction, and marker interaction patterns, NeXtMarker captures biologically meaningful signal in a fully self-supervised manner. Our experiments demonstrated that NeXtMarker resolves intercellular expression patterns, enables expert-driven phenotyping in real-world datasets, and generalizes across tissues and diseases without requiring retraining. The framework achieves high agreement with conventional integrated expression methods, while avoiding the information loss and biases introduced by background correction. Importantly, NeXtMarker does not assign cell type labels. Instead, it provides interpretable marker co-expression patterns and embedding spaces that experts can use to explore, annotate, or discover cellular phenotypes. By enabling unbiased exploration of multiplex imaging data, NeXtMarker offers a robust and scalable foundation for spatial biology analysis in both well-characterized and novel tissue settings. A current limitation is the reliance on approximate cell centers for

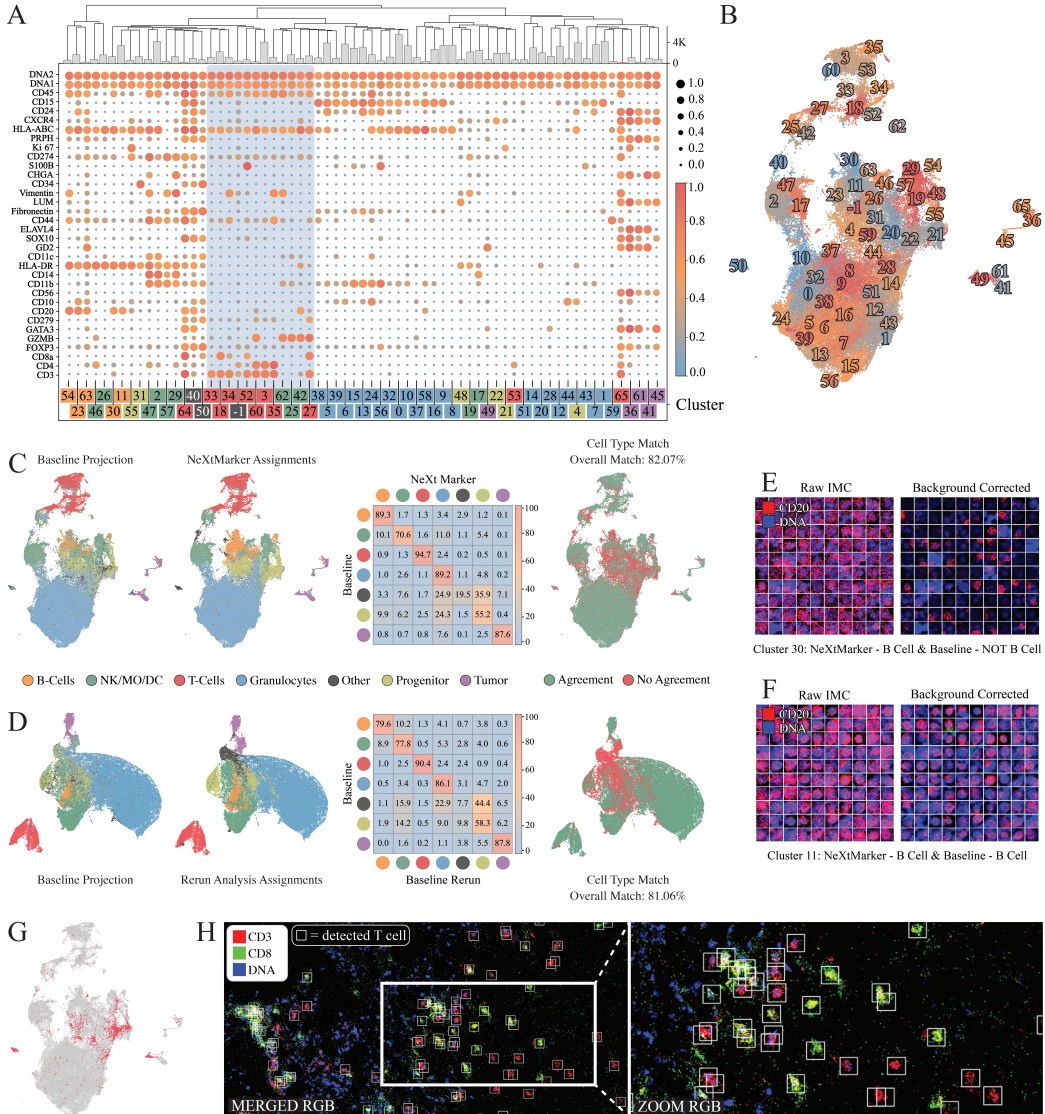

Figure 3: **A**: Cluster-wise average marker activation from NeXtMarker's interpretable features with corresponding cluster sizes. **B**: UMAP visualization of cluster assignments. **C**: Cell-type mapping from IE baseline and NeXtMarker projected onto the same UMAP. **D**:Cell-type mapping from IE baseline and IE rerun projected onto the same UMAP. **E**: Cells identified as B cells by NeXtMarker but not by the baseline analysis (cluster 30). **F**: Cells identified as B cells by both NeXtMarker and the baseline analysis (cluster 11). **G**: cells annotated as 'other' from the IE baseline in the NeXtMarker feature space. **H**: Osteosarcoma IMC image with CD3, CD8 and DNA marker with boxes indicating identified T cells.

patch extraction. Future work could integrate detection directly into the model or extend NeXtMarker to process unstructured image regions without localization.

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
