# OpenReview forum: "NeXtMarker: Contrastive Learning for Marker-Level Interpretability in Single-Cell Multiplex Imaging"
_MICCAI.org/2025/Workshop/COMPAYL — COMPAYL 2025_

### Official Review · Reviewer_TrxP · 2025-07-11
**Detailed manuscript with clear explanation of the logic behind the technical aspects**

**Rating:** 5
**Confidence:** 3

**Review:**

Summary
The authors introduced a self-supervised method for single-cell analysis of multiplex images that uses a similar framework to SimCLR, but extended for spatial biology use-case. It is a solution that highly differs from traditional methods of integrated expression quantification, in terms of not requiring cell segmentation maps and that it learns features specific to a single marker. It is a clear, well-thought and logical solution that is elaborately detailed in both the technical setup and validation. The solution also supports interpretability of results, which is important for relating back to biological insight. This work is significant as spatial biology images are challenging to analyse due to complexity, high dimensionality and expert knowledge is required across the fields of biology and data science. The use of a self-supervised method could greatly increase the accessibility of spatial biology analysis to more professionals.

Strengths:
NeXtMarker is highly automated, thus it reduces dependence on user inputs, which can be labour intensive. Key examples include the elimination of cell segmentation and manual pre-processing, both of which require time and biological knowledge for result evaluation.

Interpretable feature extraction is unique as features specific to each biological marker is learned, hence the marker that contributes to a given phenotype is traceable.

Weaknesses:
Did not compare in-depth to specific existing image analysis tools (e.g., MCMICRO, CellProfiler, etc). Authors only mentioned integrated expression as a whole and briefly talked about Steinbock.

---

### Official Review · Reviewer_K2ng · 2025-07-14
**A deep learning framework for single cell level analysis of multiplex images**

**Rating:** 2
**Confidence:** 3

**Review:**

This paper presents a deep learning framework for single cell level analysis of multiplex images. A key feature of the method is marker specific normalization, but it is not clear how this has been derived or tested compared to existing approaches e.g., https://link.springer.com/article/10.1186/s13040-016-0088-2. Similarly, results have been presented for various steps in the pipeline without comparing with an existing method. I am also not clear on segmentation performance either as it is not compared to state-of-the-art and no evaluation metrics are presented. This pipeline might be the best, but it is not clear as the results are not compared to benchmarks.